# Ultra-deep sequencing validates safety of CRISPR/Cas9 genome editing in human hematopoietic stem and progenitor cells

M. Kyle Cromer [1,2,4], Valentin V. Barsan[2,4], Erich Jaeger [3], Mengchi Wang[3], Jessica P. Hampton [2], Feng Chen[3], Drew Kennedy[3], Jenny Xiao[3], Irina Khrebtukova[3], Ana Granat[3], Tiffany Truong [3] & Matthew H. Porteus [2] ✉

As CRISPR-based therapies enter the clinic, evaluation of safety remains a critical and active area of study. Here, we employ a clinical next generation sequencing (NGS) workflow to achieve high sequencing depth and detect ultra-low frequency variants across exons of genes associated with cancer, all exons, and genome wide. In three separate primary human hematopoietic stem and progenitor cell (HSPC) donors assessed in technical triplicates, we electroporated high-fidelity Cas9 protein targeted to three loci (AAVS1, *HBB*, and *ZFPM2*) and harvested genomic DNA at days 4 and 10. Our results demonstrate that clinically relevant delivery of high-fidelity Cas9 to primary HSPCs and ex vivo culture up to 10 days does not introduce or enrich for tumorigenic variants and that even a single SNP in a gRNA spacer sequence is sufficient to eliminate Cas9 off-target activity in primary, repair-competent human HSPCs.

The CRISPR system, consisting of a CRISPR/Cas protein coupled with a guide RNA (gRNA), has demonstrated remarkable versatility for site-specific genome editing. To ensure safe clinical translation of CRISPR systems for genome editing, insertions and deletions (indels) should occur only at the intended genomic site without off-target effects, through either non-homologous end joining (NHEJ) or homology-directed repair (HDR) pathways. Unintended genome editing can occur with low-fidelity Cas enzymes or when the gRNA directs cleavage to sequences similar to the target sequence, leading to the incorporation of off-target mutations that may have oncogenic or otherwise deleterious consequences. Detecting genotoxicity of genome editing methods remains fundamental to safe clinical implementation.

Several recent reports have shown that DNA double-strand breaks (DSBs) introduced by Cas9 initiate a p53 response in pluripotent and cancer cell lines that results in cell cycle arrest and/or apoptosis[1,2]. Because cells with loss-of-function mutations in p53 do not suffer the same degree of toxicity following genome editing and may proceed through the cell cycle with unresolved DSBs, these studies suggested that Cas9-mediated cleavage can enrich for p53 mutations. However, the findings from these studies depend on the presence of p53 mutations in the initial pool of cells prior to (not as a consequence of) Cas9 delivery, which would not be expected to occur in primary cells derived from healthy donors. These studies were also conducted in immortalized cell lines that typically have gross chromosomal abnormalities (polyploidy, aneuploidy, translocations, etc.) with dysfunctional DNA damage and nucleic acid delivery-sensing responses and used stable expression of Cas9 that contributes to prolonged cellular stress[3-5].

Significant efforts have thus been directed at not only predicting possible off-target genomic coordinates a priori[6-8], but also toward the development of empirical lab-based methods for detecting sites of off-target activity following genome editing[9-13]. While these experimental methods report activity at many candidate sites, which may be missed by in silico prediction methods, there remains concern that such techniques depart from clinical protocols for Cas9 delivery as these methods typically involve constitutive expression of wild-type Cas9 in immortalized cell lines or Cas9 delivery to cell-free genomic DNA (gDNA). Wet lab-based methods may thus have a high false positive

[1]Department of Surgery, University of California, San Francisco, San Francisco, CA, USA. [2]Department of Pediatrics, Stanford University, Stanford, CA, USA. [3]Illumina, San Diego, CA, USA. [4]These authors contributed equally: M. Kyle Cromer, Valentin V. Barsan. ✉e-mail: mporteus@stanford.edu

rate in a clinical situation. Therefore, there is a significant need to assess the performance of prediction algorithms and empirical methods in more therapeutically relevant contexts (i.e., via transient ribonucleoprotein (RNP)-based delivery of high-fidelity Cas9[14] to human primary cells ex vivo).

The importance of long-term safety of genome editing/gene therapy in the clinic was illustrated recently when two sickle cell disease gene therapy trials (NCT02140554 and NCT04293185) were paused after two patients developed myeloid malignancies from either cytotoxic conditioning chemotherapy or insertional mutagenesis of the lentiviral vector[15]. Because of these safety concerns, in this study we sought to determine if oncogenic variants are introduced during Cas9 editing and/or the ex vivo expansion workflow. The ideal methodology necessitates ultra-deep sequencing since mutations with a variant allele frequency (VAF) below 1% remain undetected by most genome-wide off-target detection techniques. This is in part because the signature of Cas9 nuclease activity is a spectrum of indels rather than primarily single nucleotide variants (SNVs). Therefore, an ultra-deep sequencing workflow capable of detecting SNVs as well as indels, amplifications, and multi-nucleotide variants (MNVs) has the potential to dramatically increase sensitivity for detection of the full spectrum of oncogenic off-target editing activity from 1% to <0.1% VAF, which will be necessary to identify low frequency variants that could initiate pathogenic clonal expansion.

To achieve such high sensitivity for clinical implementation, we developed a NGS workflow which allows identification of low-frequency events at the most important genomic regions for assessing high-risk genotoxicity: exons of genes associated with cancer that are routinely screened for mutations in existing clinical oncology workflows. We found no evidence that clinical Cas9 ex vivo genome editing in multiple primary hematopoietic stem and progenitor cell (HSPC) donors at three separate genomic loci introduces or enriches for oncogenic mutations. These findings were confirmed by whole-exome sequencing (WES) and whole-genome sequencing (WGS) when targeting AAVS1. Importantly, our workflow was not dependent on guide RNA homology and therefore was unbiased and highly scalable. We have therefore defined a method, adapted from existing clinical oncology NGS diagnostics, that can be used to evaluate the safety profile of current and next-generation genome editing tools, ex vivo culture protocols, and cell-based products prior to clinical translation.

## Results

### Sequencing pipeline attains high coverage of tumor suppressors and oncogenes

To perform ultra-deep sequencing of tumor suppressors and oncogenes, we used a hybrid-capture NGS assay for detection of DNA variants at high depth across the exons of 523 cancer-relevant genes (spanning 1.94 Mb of DNA) using unique molecular indexes (leveraging the TruSight Oncology 500 kit)[16]. These 523 genes comprise known oncogenes in key guidelines of the most common cancer types as well as many recurring genes that are commonly mutated in cancer and under clinical investigation, spanning from non-small cell lung cancer to pancreatic adenocarcinoma (Supplementary Table 1). Prior work has shown a high degree of concordance (both positive and negative agreement) between the TSO500 panel and whole exome sequencing (WES) for measurement of mutation burden (nonsynonymous mutations per kilobase of DNA)[16]. Because the panel is derived from the genetics of human cancer and was not custom-built according to gRNA design, it is unbiased with respect to the genome editing process and focuses on the highest risk regions of the genome in terms of assessing potential oncogenicity of a genome editing strategy. The commercial availability of the TSO500 panel also makes it accessible to the broader genome editing community and, especially, clinical laboratories.

HSPCs from three separate healthy donors were subject to four conditions: Mock electroporated as well as three different Cas9

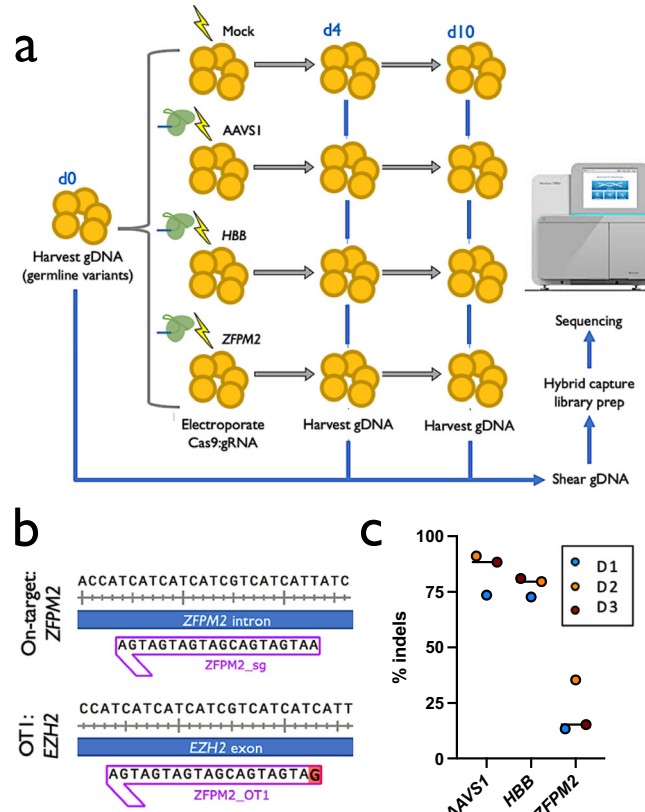

**Fig. 1 | Experimental design and confirmation of on-target activity.**
**a** Experimental design: CD34[+] HSPCs from 3 donors underwent gDNA harvesting at d0 (to establish germline variants) and were then subject to mock electroporation or Cas9 treatments with gRNAs corresponding to sites at AAVS1, HBB, and ZFPM2. Cells were cultured and gDNA was harvested again at d4 and 10 post-editing. **b** Predicted off-target cut site (OT1) of ZFPM2 guide in exon 5 of the EZH2 gene, based on sequence homology. Mismatch in gRNA is shown in red. **c** On-target activity of AAVS1, HBB, and ZFPM2 gRNAs determined by PCR amplification of the genomic region surrounding the predicted cut sites followed by Sanger sequencing and analysis of indels by TIDE 4 days post-editing. Bars represent median. N = 3 separate HSPC donors.

treatments with gRNAs targeting sites at AAVS1, HBB, or ZFPM2 (Fig. 1a; Supplementary Table 2). Cas9 activity at AAVS1 and HBB have been extensively documented in the literature and these sites were chosen due to their relatively high- and low off-target activity, respectively[10,11]. Notably, the HBB gRNA used here is currently in phase I clinical trials for correction of the single SNP responsible for sickle cell disease[17,18]. As a positive control that we expected to elicit off-target activity in the TSO500 panel, we designed a gRNA targeting intron 3 of ZFPM2, which has a predicted off-target site in exon 5 of EZH2. This off-target site differs by a single nucleotide at position 1 of the spacer sequence, the site furthest from the PAM that has the least bearing on Cas9 specificity (Fig. 1b), and is the highest ranked off-target site for the ZFPM2 guide by COSMID[6]. EZH2 was chosen as a relevant positive control because of its well-characterized role in a wide range of tumor types[19,20] and the relevance of both loss- and gain-of-function mutations in myelodysplasias[21–24], making it especially relevant for HSPC editing.

The methodology used to assess Cas9 activity following genome editing was adapted from a clinical formalin-fixed paraffin-embedded (FFPE) tissue workflow for genomic (gDNA) harvested from primary CD34[+]-purified umbilical cord blood-derived HSPCs across three separate healthy donors. Frozen cells were thawed and expanded for 2 days in HSPC media at 100 K cells/mL and then targeted in the four treatment groups ($2-5 \times 10^5$ cells per treatment group) as reported

previously[17,18,25,26] (Fig. 1a). Genomic DNA was harvested from $3–4 \times 10^5$ cells at day 0 to establish germline variants and then cells were split into treatment groups, electroporated, and re-plated in fresh media. Because prior reports have shown that indel formation saturates 4 days after electroporation of HSPCs with Cas9 RNP[26], we harvested $4 \times 10^5$ cells from each treatment group at day 4 and extracted gDNA for analysis. To determine whether enrichment of tumorigenic variants was occurring in our ex vivo-expanded HSPC populations, as well as to gain insight into whether ex vivo expansion itself (independent of Cas9 activity) was enriching for tumorigenic variants, we also harvested gDNA from the remaining cells at 10 days post-targeting.

To ensure that high levels of on-target activity occurred for each gRNA, we performed targeted PCR amplification of the genomic region surrounding the predicted cut site followed by Sanger sequencing and analysis of indels by TIDE[27]. A high frequency of on-target indels were observed across all three donors for AAVS1 and *HBB* gRNAs (Fig. 1c). While consistent across all donors, the *ZFPM2* gRNA induced fewer indels, which was expected owing to its high degree of predicted off-target activity as well as the fact that this guide was not screened for efficiency, in contrast to previously optimized AAVS1 and *HBB* gRNAs.

First, pilot experiments were performed to confirm that the TSO500 pipeline could be adapted from FFPE-derived tissue to gDNA harvested from primary cells. To determine the optimal amount of DNA for application to the sequencing pipeline, a range of 10–30 ng of DNA was used as input for library preparation using the hybrid capture-based TSO500 Library Preparation Kit. Reads were mapped to the human genome (build hg19) and raw sequencing data was processed through a custom bioinformatic pipeline (Supplementary Fig. 1A) to identify indels, SNVs, and MNVs. Pilot experiments confirmed successful adaptation of the sequencing pipeline to gDNA harvested from primary cells in culture, and that at least 30 ng of input DNA was necessary to achieve a median exon coverage (MEC) of 2000 (Supplementary Fig. 1B). To simultaneously detect intended edits, we supplemented the TSO500 panel with probes specific to the regions targeted by AAVS1, *HBB*, and *ZFPM2* gRNAs (Supplementary Table 3).

Following initial pilot experiments, raw sequencing data yielded a mean MEC > 3550 for all samples per technical replicate, corresponding to a minimum limit of detection (LoD) and sensitivity of 0.205% and 95%, respectively (Fig. 2a; Supplementary Fig. 2). Moreover, because three technical replicates were sequenced for almost all timepoints and conditions, which are factored into the mean MEC of >3550, our LoD in these samples was further pushed to a limit of <0.07% VAF. Variants were consistent across technical replicates in terms of the types of variants called, with no significant differences comparing Mock to Cas9 treatments (Supplementary Fig. 3). We also observed a high degree of concordance across technical replicates, with a median of 98.31% of variants called in all replicates for each treatment for each donor (Fig. 2b; Supplementary Table 4). These data indicated that the total number of variants across replicates was more dependent on donor than either time in culture or treatment with Cas9. In addition, the number of variants within each donor did not consistently increase due to time in culture (i.e., day 0 v. day 4 v. day 10) or treatment with Cas9. Consistent with these results, we found that read depth across the genome was more heavily influenced by donor than any other factor (Supplementary Fig. 4). In addition, while chromothripsis was recently reported as a rare consequence of on-target Cas9 cleavage[28], we found no apparent drop in read depth across our bulk population of HSPCs in variants proximal to the intended cut site for any Cas9 treatment.

## Few variants found in treatments after filtering non-pathogenic germline mutations

To gain insight into the characteristics of the variants identified in our cohort, we plotted VAF by MEC for Mock samples at days 0, 4, and 10 (Supplementary Fig. 5A). Strikingly for all donors across all timepoints, the VAF frequencies trended toward 0.5 and 1.0 as MEC increased, which correspond to heterozygous and homozygous germline variants, respectively. Because all variants were found within a panel of tumor suppressors and oncogenes, yet all HSPCs were derived from normal, healthy donors, we expected virtually all variants identified in Day 0 and Mock conditions to be non-pathogenic. Indeed, when filtering out both synonymous variants as well as those previously reported to occur >10 times in comprehensive germline databases[29,30], only a handful of variants remained (a mean of 3.9 variants remaining from 1490.1 reproducible variants per condition). Again, we found no consistent increase in the number of variants as HSPCs were cultured from d0 through d10. Interestingly, we observed several consistent variants that, while present in our germline database and consequently filtered, were found at intermediate VAFs rather than trending toward 0.5 or 1.0. While these variants were consistent within, but not across donors, none of these were found in the exons of genes associated with clonal hematopoiesis of indeterminate potential[31,32] (note: the age of the donors for the source of the HSPCs is not known). Therefore, we believe these mutations represent either sequencing artifacts or bona fide HSPC donor chimerism that occurred prior to ex vivo culture or editing.

To determine whether editing with Cas9 introduces variants in tumor suppressors or oncogenes, we then plotted VAF x MEC for all Cas9 treatment groups at days 4 and 10 for all three donors (Fig. 2c; Supplementary Fig. 5B). Again, as expected for heterozygous and homozygous germline mutations, unfiltered variants trended toward VAFs of 0.5 and 1.0 as MEC increased. We next filtered out non-pathogenic variants by eliminating all called mutations that are synonymous and/or have been previously reported in the germline variant database. We found that our Cas9 treatments had fewer variants remaining than our Mock conditions (a mean of 3.3 variants remaining from 1487.9 reproducible variants per condition). Because any variants also found in Mock samples would not have been introduced by Cas9, we removed these for downstream analyses (Fig. 3a). Of eighteen Cas9 treatments, only six variants remained after filtering, and four of these were the expected *EZH2* mutations in Donors 2 and 3 within both day 4 and day 10 *ZFPM2* treatments. The other two variants that remained after filtering germline, synonymous, and Mock mutations were both SNVs found in d10 *ZFPM2* treatments in Donors 2 and 3 at <0.0015 VAF, which approaches our limit of detection. It is important to note that while only six variants remained in our treatment groups, the filters we applied to our Cas9 conditions were fairly permissive. Because Cas9 introduces indels far more frequently than SNVs at sites that display homology to the gRNA, if we applied additional filters to our variants (i.e., removed SNVs as well as sites with no homology to the gRNA), only *EZH2* mutations would remain.

## *EZH2* off-target activity eliminated by homozygous SNP in Cas9 gRNA spacer

In Donors 2 and 3, the expected *EZH2* off-target site displayed the highest VAF in both d4 and d10 timepoints across all three replicates at high confidence (3,893x coverage) at an average of 19.3% off-target activity (Fig. 3a). Interestingly, the *EZH2* VAF in these donors decreased from day 4 to day 10 (mean of 21.7% to 16.9%, respectively), perhaps the result of a selective disadvantage for cells that harbor indels in this gene. The indel spectrum within *EZH2* was characterized (Supplementary Fig. 6) and total frequency was validated by PCR amplification, Sanger sequencing, and analysis of indels by TIDE (Fig. 3b). Notably, even without filtering Mock variants, mutations in *EZH2* comprised the majority of calls in Cas9+*ZFPM2* gRNA treatments in Donors 2 and 3 (Fig. 3c; Supplementary Fig. 7). Surprisingly, we found no detectable off-target activity at *EZH2* in Donor 1 by either NGS or TIDE (Fig. 3b) despite a

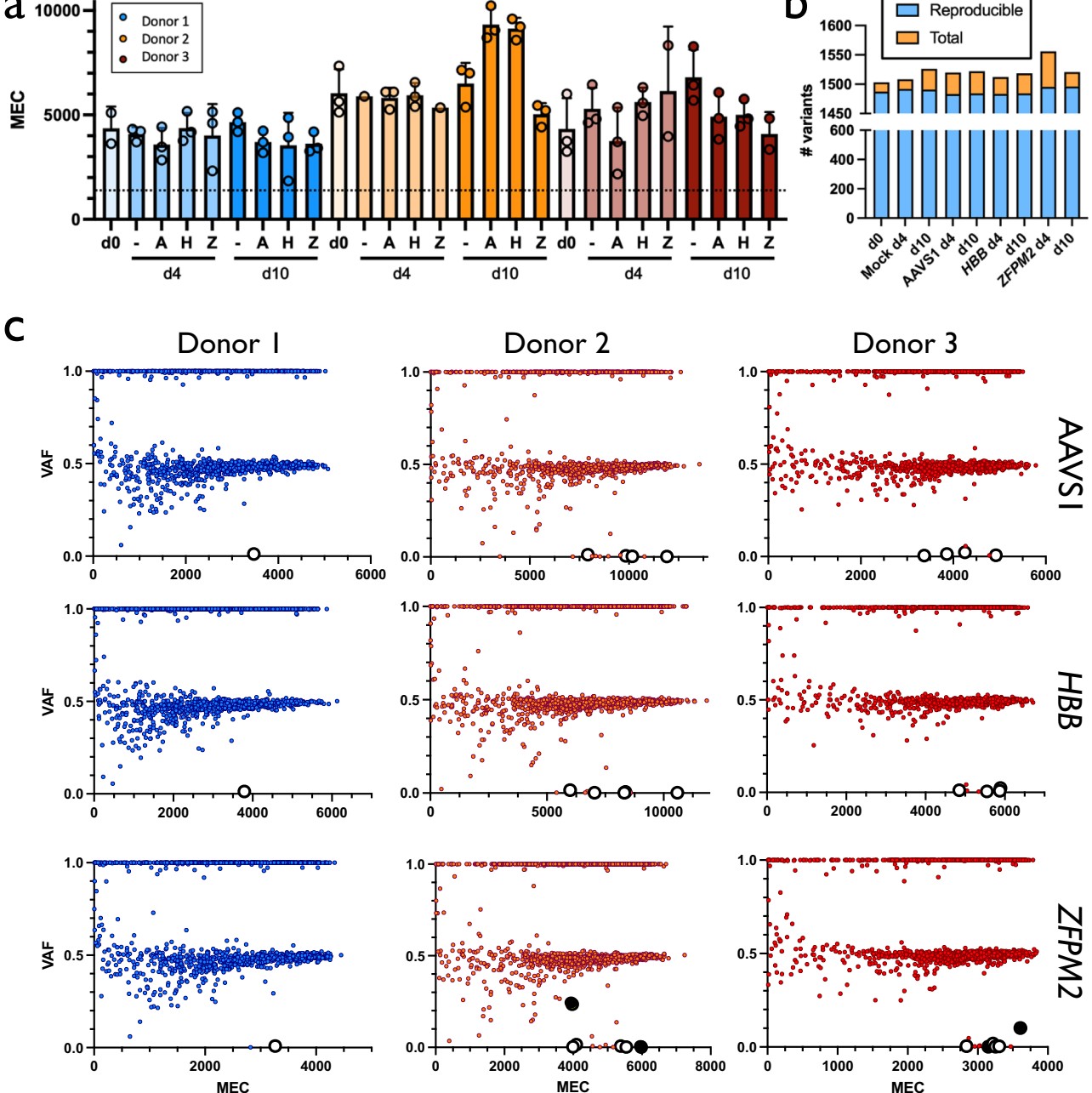

**Fig. 2 | Summary of TSO500 sequencing data. a** MEC for each treatment for each donor. Treatments are Mock electroporated (-), AAVS1- (A), *HBB*- (H), and *ZFPM2*-targeted (Z). Individual points represent technical replicates. Columns and error bars represent mean and standard deviation. Dotted line indicates recommended lower specification limit, set at 1300 MEC. **b** Number of reproducible variants across technical replicates from total called by treatment group. Columns represent mean variants called for the three donors within each treatment. **c** VAF x MEC for all variants found among technical replicates for Cas9 treatments for each donor at d10. Large white points are those that remained after removing germline and synonymous variants. Large black points are those that remain after removing variants present in Mock within each donor.

high degree of on-target activity at *ZFPM2*. Upon investigation of the Sanger trace at this site in Donor 1, we found a homozygous SNP at position 6 of the spacer sequence (Fig. 3d). Due to the specificity of high-fidelity Cas9, which has been reported to reliably reduce off-target activity by 20-fold[14], it is likely that this homozygous SNP eliminated all activity at this site (below the detection threshold of the TSO500 panel). The exceptional specificity of high-fidelity Cas9 protein, when transiently delivered to primary cells, is evident from the single SNP outside of the core region of the spacer sequence in Donor 1 that was sufficient to eliminate all detectable activity at this site.

**Whole exome sequencing confirms absence of off-target activity from ex vivo culture and genome editing**

Because transient delivery of Cas9 RNP and up to 10 days of ex vivo culture elicited few variants in the TSO500 panel, we next sought to expand our search for off-target activity to the entire exome. To do so, we electroporated high-fidelity Cas9 pre-complexed with AAVS1 gRNA to a single HSPC donor. We used the AAVS1 gRNA because it has been described as less specific than the *HBB* gRNA and we wanted to increase the chances of detecting any exonic off-target site. We then harvested gDNA from AAVS1-targeted and Mock electroporated treatments at d10 post-editing and subjected both samples to an exome capture

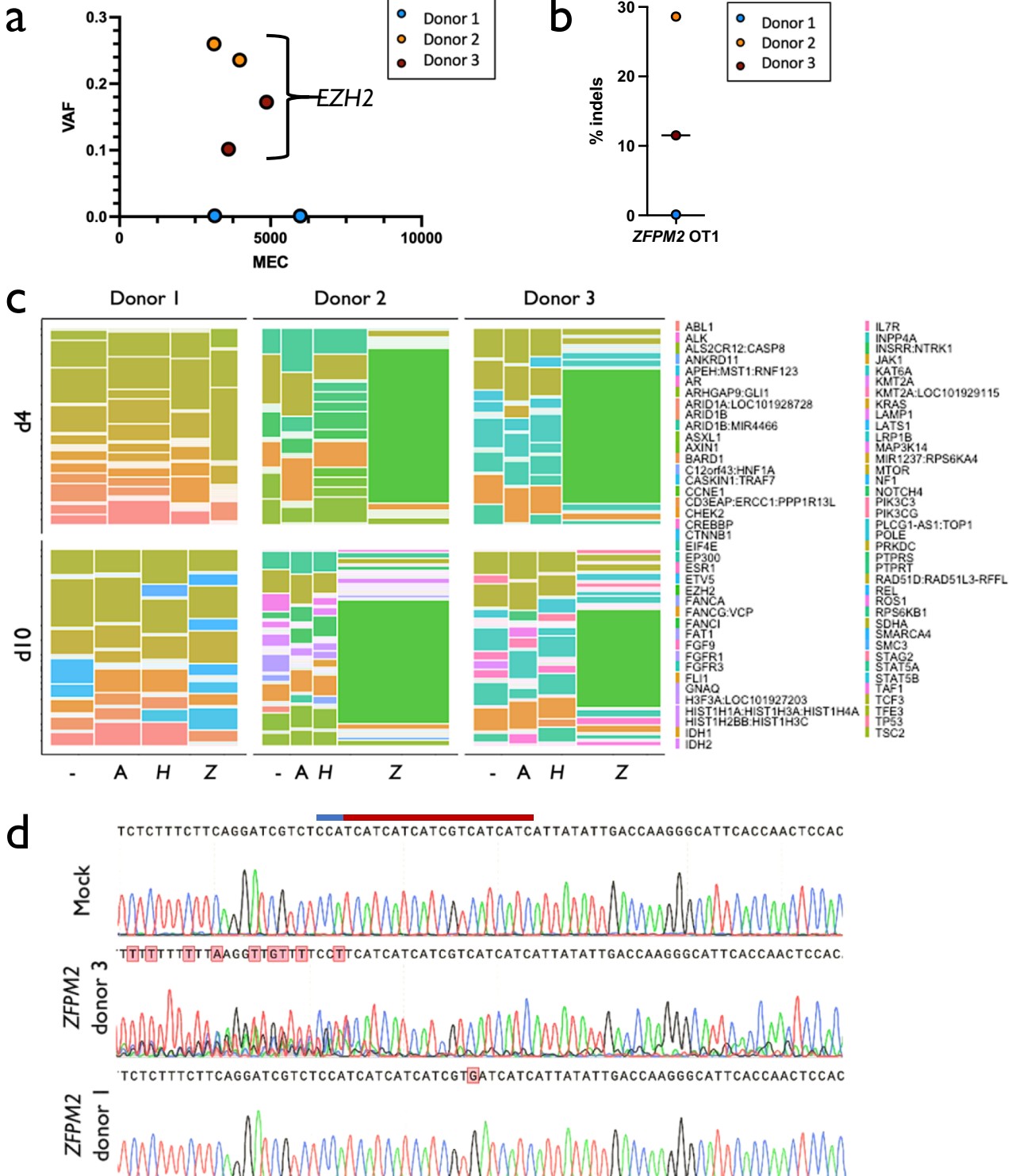

**Fig. 3 | Variants identified in Cas9 treatments. a** VAF x MEC for variants remaining from all Cas9 treatments after removal of synonymous, germline, and Mock calls. Donor 1 had no variants remaining after filtering. **b** Percent indels in *EZH2* identified by PCR amplification, Sanger sequencing, and TIDE analysis using d4 gDNA. Donor 1 had no detectable activity. **c** Mosaic plot of genes harboring mutations within each donor and Cas9 treatment at d4 and 10. Area is proportional to the number of times variants were called in a particular gene within a particular treatment group. Filtering removed germline and synonymous variants. For each donor and timepoint, conditions are ordered as Mock, AAVS1, *HBB*, and *ZFPM2* (-, A, *H*, and *Z*, respectively). **d** Sanger chromatograms at predicted *EZH2* off-target site. The PAM site and spacer are depicted as blue and red lines, respectively. Homozygous SNP in Donor 1 abrogated detectable editing activity.

panel and NGS. This achieved high read depth across our target regions; mean alignment coverage at 1988 and 2054 for AAVS1 and Mock treatments, respectively (Supplementary Table 5). Prior to filtering, we identified 38,431 and 38,527 variants in Mock and AAVS1

treatments, respectively (Fig. 4a). To identify variants that may have resulted from Cas9 treatment, a tumor-normal pipeline was used to call somatic variants that were unique to the AAVS1 treatment ("tumor") after subtracting the Mock as background ("normal"). In addition, we

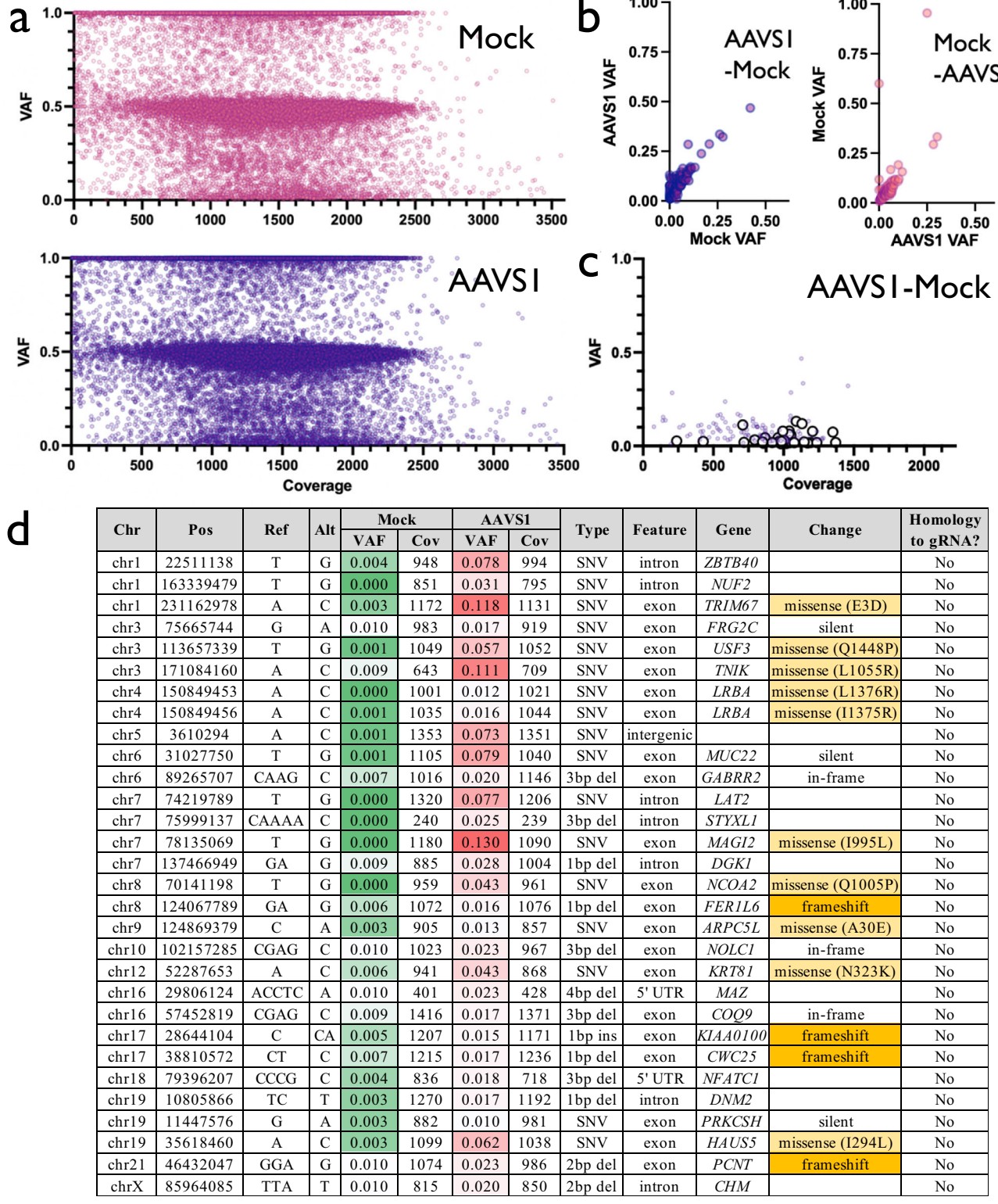

**Fig. 4 | Variants identified by whole-exome sequencing. a** VAF x Coverage for all variants called by exome sequencing pipeline in Mock and AAVS1 d10 conditions. **b** VAF for variants called by tumor-normal pipeline when AAVS1 is used as tumor and Mock as normal inputs (left panel), and when Mock is used as tumor and AAVS1 as normal (right panel). **c** VAF x Coverage for 137 variants shown in **b**. 30 large white points are those that remained after removing variants with Mock VAF > 0.01. **d** Annotation for all 30 AAVS1 variants. Homology to gRNA is defined as 10 or more matches to spacer+PAM within 20 bp upstream or downstream of variant.

inverted the tumor-normal designation (i.e., treating Mock as tumor and AAVS1 treatment as normal) to estimate our background frequency of somatic calls resulting from this pipeline. These analyses identified 137 somatic variants in the AAVS1 treatment and 92 variants in the Mock condition (Fig. 4b). Because this pipeline is typically used

to identify somatic variants in heterogeneous tumor samples, any mutation with a VAF notably greater than the "normal" sample was flagged. However, no off-target mutation introduced by Cas9 would have been present at any detectable VAF in the Mock condition. Therefore, we removed variants found at >0.01 VAF in the Mock

treatment, leaving 30 somatic mutations for further analysis (Fig. 4c). Though Cas9 nuclease activity typically introduces indels surrounding sites with a high degree of homology to the spacer sequence, most remaining variants after filtering (17 of 30) were SNVs and none of the 30 mutations were found to have >10 bp match to the spacer + PAM sequence within 20 bp upstream or downstream of the called variant (Fig. 4d). Therefore, neither targeted tumor suppressor/oncogene sequencing nor WES was able to identify any somatic mutations that occurred because of Cas9 activity. We believe that the variants identified as somatic mutations in both Mock and AAVS1 treatments represent either real variation that occurred over the course of the 10-day ex vivo HSPC expansion or are sequencing artifacts.

### Whole-genome sequencing confirms absence of off-target activity from ex vivo culture and genome editing

While intronic and intergenic variants are associated with disease progression, these are often excluded from clinical genetic workflows due to (1) difficulty in interpretation, (2) limited clinical actionability of non-coding variants, and (3) inability to achieve equivalent high coverage across most of the non-coding genome compared to that possible with exon capture or other targeted arrays. Nevertheless, to determine whether ex vivo CRISPR-mediated genome editing introduced variants that may have been missed by both the TSO500 panel as well as whole-exome sequencing, we performed whole-genome sequencing on gDNA harvested from one HSPC donor at d3 post-editing with our AAVS1 gRNA (as well as Mock sample for input control). This achieved modest read depth across our target regions; mean alignment coverage at 184 and 212 for AAVS1 and Mock treatments, respectively (Supplementary Fig. 8A, B; Supplementary Table 6). Using an established bioinformatic pipeline, we then called somatic variants present in the AAVS1 sample that were not found in the Mock control, identifying a total of 26,673 variants (mean 154 reads across called variants) (Fig. 5a, b). When plotting VAF vs. coverage depth, we observed that virtually all high-VAF variants were found at low read depth, indicating that these are likely not real variants unique to AAVS1 treatment but rather germline variants that were not sequenced at sufficient depth to be detected in the Mock sample as well. This hypothesis was supported by plotting coverage in AAVS1 vs. coverage in Mock treatment for all called variants, which revealed a linear correlation between coverage across both Mock and AAVS1 samples (i.e., high-VAF, low coverage variants in AAVS1 treatments were likely to be covered at low read depth in Mock as well) (Fig. 5d). We observed a clustering of low-coverage variants at VAFs of 1 and 0.5 (Fig. 5a, b), reinforcing the notion that these are germline homozygous or heterozygous variants that were also present in the Mock treatment but not called due to low coverage. Next, to remove genotype background variants, we subtracted those detected in the Mock condition from AAVS1 variants (Fig. 5c) and again found that virtually all high VAF variants were sequenced at low coverage, likely representing sequencing noise below the limit of detection. We further filtered these variants to remove calls below 0.1% VAF in the AAVS1 treatment and variants >1% VAF in the Mock sample—leaving a total 173 variants (Fig. 5e). Of these, 19 variants resided at the predicted on-target site for the AAVS1 gRNA, comprising all the highest VAF calls at coverage >100. Of remaining candidate off-target variants (Extended Data), we found no homology to the AAVS1 gRNA, likely indicating false positives due to low coverage in the AAVS1 and/or Mock treatments.

As further evidence that the mutations found in the AAVS1 treatment represent sequencing artifacts, we performed additional analysis where the Mock sample was treated as the "edited" condition and the AAVS1 sample was treated as the background control. While we expect few if any true variants to be present in the Mock that were not also present in the AAVS1 treatment, we found a total of 41,200 "somatic" variants in the Mock (mean 174 reads across called variants) (Fig. 5f and Supplementary Fig. 8C, D). This list was further reduced to 171 variants

after performing analogous filtering as before (Supplementary Fig. 8E, F). Because a nearly identical number of mutations were found in the Mock as in the AAVS1 treatment, as well as the lack of gRNA homology to any of off-target sites in the AAVS1 treatment, we believe that WGS data indicates no evidence of additional (intronic or intergenic) off-target variants introduced by CRISPR-mediated targeting at AAVS1 at the given coverage depth.

## Discussion

The first CRISPR-based therapies have entered early human clinical trials and many others are entering drug development pipelines[33]. There is a growing need to establish the long-term safety of edited human cells (ex vivo and in vivo) by CRISPR nucleases and vectors. We have defined a method to evaluate the safety profile of genome editing tools, ex vivo culture protocols, and cell-based products prior to clinical translation. Establishing the appropriate metrics for assessing genomic stability after genome editing continues to be an important and active area of study. For instance, whereas WGS is the only way to capture variants/abnormalities across the entire genome, read depth per base pair to achieve high sensitivity required for genome editing purposes across a population of cells remains cost-prohibitive and technically complex. Sequencing the entire human genome of bulk cells only allows detection of high frequency events due to low per-base coverage. Alternatively, limiting sequencing to the most conserved and functionally significant regions of the genome (i.e., exons, which comprise 1% of the genome[34]) allows for greater coverage and therefore greater detection power of lower frequency variants. While NGS can identify somatic mutations introduced by ex vivo culture and genome editing methods, sequencing breadth versus depth tradeoffs exist for scalable clinical implementation that reproducibly validates the safety of engineered cell therapies. Because cancer can occur due to expansion of even a single mutated clone, in this study we applied this concept to further limit sequencing to exons of the most common tumor suppressors and oncogenes, identified in an unbiased way with respect to genome editing from cancer genetics, to detect extremely low frequency events.

Cas9 can initiate DNA DSBs at both on- and off-target sites, potentially leading to unintended genomic abnormalities. In fact, several studies reported the enrichment of p53-inactivating mutations following CRISPR-based editing in immortalized human cell lines when a subset of p53 mutant cells were spiked into the initial pool of cells—a crucial point as these were not mutations initiated by Cas9 editing[1,2]. This is reinforced by prior studies in human primary cells which found that Cas9 RNP delivery did not introduce mutations in p53 or 129 other cancer-related genes (using the Stanford Solid Tumor Actionable Mutation Panel), though at a limit of detection of 5% VAF[35,36]. Therefore, we developed a tumor suppressor/oncogene ultra-deep sequencing pipeline to determine whether editing and short-term ex vivo expansion leads to disruption and/or enrichment of cancer-associated variants when delivered in a clinically relevant context—i.e., when high-fidelity Cas9[14] is transiently delivered as RNP via electroporation to human primary HSPCs without subpopulations of cells with pre-existing tumorigenic variants. Toward this end, our workflow interrogated the exons of 523 known tumor suppressors and oncogenes and achieved levels of detection of germline and somatic mutations at <0.1% VAF, representing a far lower limit of detection (>50-fold) than achieved in prior studies.

When editing with three separate gRNAs (targeting AAVS1, HBB, and ZFPM2), ultra-deep sequencing of >500 tumor suppressors and oncogenes found no detectable variants (>0.002 VAF) that could be attributed to Cas9 activity or ex vivo expansion (aside from the expected EZH2 off-target site in the ZFPM2 treatment group). These findings were further confirmed by the absence of any off-target activity at sites resembling the AAVS1 gRNA by WES or WGS. In this clinically relevant context, transiently delivered high-fidelity Cas9 RNP

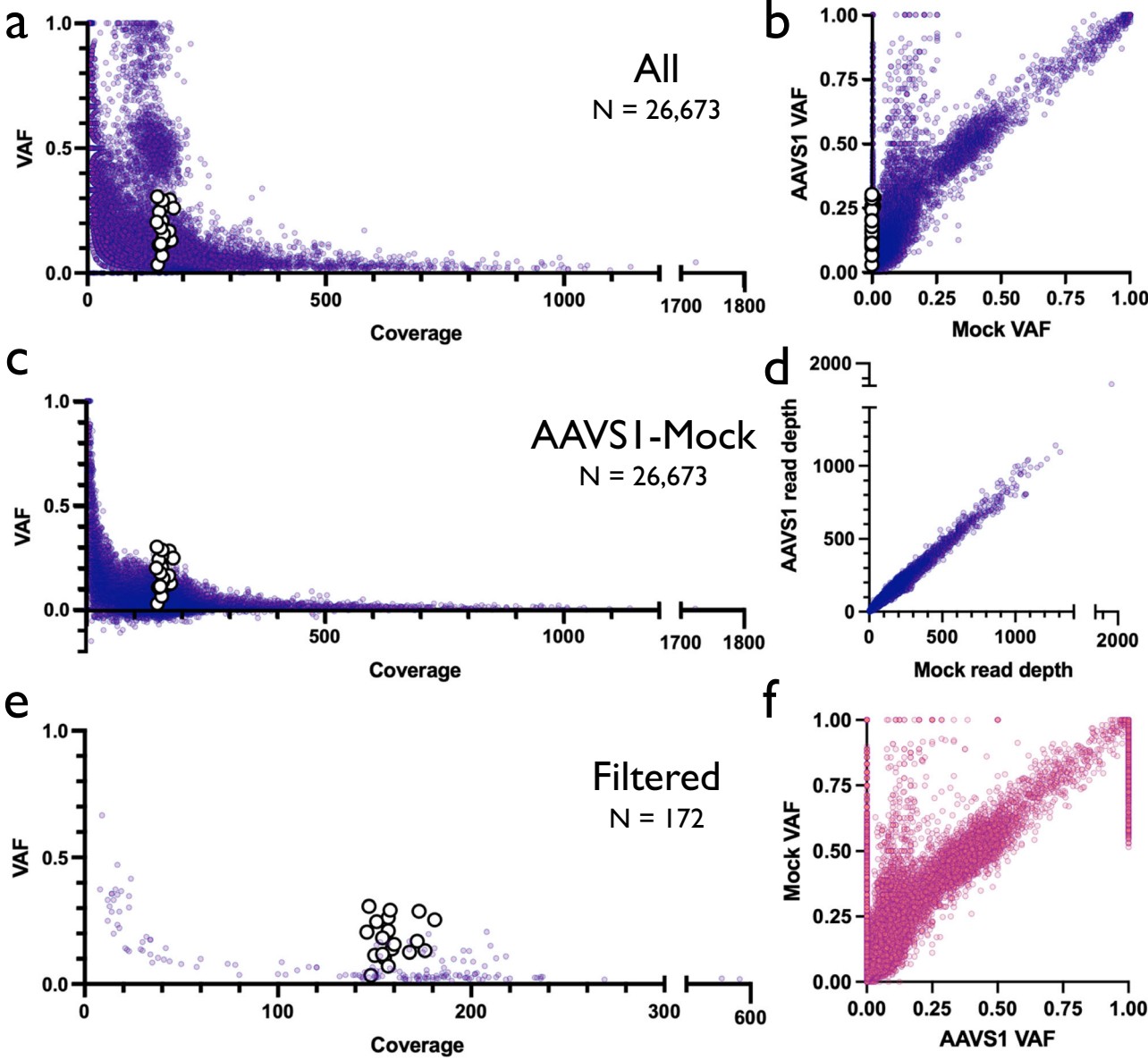

**Fig. 5 | Variants identified by whole-genome sequencing. a** VAF x Coverage for all variants called by WGS pipeline at d3 in AAVS1 treatment with Mock as background input. Large white points depict on-target AAVS1 variants. **b** AAVS1 VAF x Mock VAF for all called variants in AAVS1 treatment Large white points indicate on-target AAVS1 variants. **c** VAF x Coverage for all variants depicted in **a** after subtraction of Mock VAF from AAVS1 VAF. Large white points depict on-target variants. **d** AAVS1 read depth x Mock read depth for all called variants in AAVS1 treatment. **e** VAF x Coverage for all variants depicted in **a** after filtering to remove calls below 0.1% VAF in the AAVS1 treatment and VAFs > 1% in the Mock sample. Large white points depict on-target variants. **f** Mock VAF x AAVS1 VAF for all called variants in Mock treatment (i.e., in Mock treatment with AAVS1 as background input).

into primary HSPCs did not introduce nor enrich tumor variants. In fact, high-fidelity Cas9 was found to be so specific that even a single homozygous SNP at position 6 of the spacer sequence was sufficient to eliminate all detectable off-target activity in *EZH2*. In light of our findings, previous reports[1,2] are likely an artifact of p53 mutant spike-in, genomic instability of cell lines, supraphysiological levels of Cas9, and/ or dramatic toxicity. Taken together, this work highlights the importance of: (1) regulating the duration and level of nuclease expression in order to limit the degree of off-target activity[11,37]; (2) minimizing toxicity through electroporation of RNP as opposed to mRNA- or plasmid-based editing[38] so that opportunities for clonal expansion are minimized; and (3) conducting experiments in the most clinically relevant models—primary human cells with functional DNA sensing and damage repair machinery—rather than immortalized cell lines with well documented genomic abnormalities[3,4].

A limitation of this work is that we only attained high coverage in the coding regions of genes, including those known to be involved in cancer. We chose this focus since off-target effects in exons, especially in tumor suppressors and oncogenes, carry the highest risk for causing adverse events and have been well described in the pathogenesis of tumors. This focus enabled high sequencing depth and, consequently, identification of extremely low frequency variants at log-fold higher sensitivity than previously reported using hybrid capture methods[35,36]. Off-target indels in non-coding regions[39] of the genome, while not part of a standard oncologic workup, were evaluated through WGS, albeit at much lower depth/sensitivity owing to cost. While our panel did include 523 cancer-relevant genes (spanning 1.94 Mb), chromosomal truncations[40,41] outside of these loci was not specifically surveyed and would also require a whole-genome approach or high-resolution array (i.e., comparative genomic hybridization) to enable detection, albeit at

lower sensitivity. We demonstrated that WGS can be used in special circumstances, although the cost and high noise of genetic variation means it is currently not practical to perform as a routine off-target evaluation. As cell and gene therapies expand in the treatment of additional diseases, an important extension of this work will be to validate our findings in additional cell types and chromatin states beyond HSPCs[42–44]. For example, we anticipate that our approach can be expanded to additional patient populations with cancer predisposition syndromes, oncogenic environmental exposures such as smoking/radiation/bone marrow stress, older age, and immunocompromised status to validate the intrinsic versus genotoxic effects of CRISPR/Cas editing.

The importance of establishing safety of cell-based therapies prior to clinical translation is illustrated by the recent development of leukemia in two patients enrolled in a lentiviral gene therapy trial for sickle cell disease, which resulted in pausing of both related trials[15]. Follow-up investigation found that leukemic cells harbored viral integrations and mutations in *RUNX1* and *PTPN11* occurred at some point during or following myeloablative conditioning and/or lentiviral integration. Disruption of both of genes have been shown to play a role in a wide variety of cancers[45–48], and due to inclusion in the TSO500 panel and the sequencing depth we achieved in this study, we would have been able to identify variants in these genes at ≥0.1% VAF prior to autologous transplantation in these trials.

In summary, we believe our study not only establishes an important benchmark for the typical degree of variation in cancer-associated genes following CRISPR-based editing and short-term ex vivo expansion, but also may become a common tool for assessing safety of cell-based products prior to transplantation (particularly in the event of clonal expansion and/or long-term ex vivo culture). Improving the ability to detect and ensure the absence of oncogenic mutations will therefore maximize the chances for successful clinical implementation and the long-term safety of site-specific genome editing therapies. Such high sensitivity safety workflows, like the one outlined here, are invaluable to ensure that the safety of CRISPR-based approaches keeps pace with the efficacy of these treatments as increasing numbers of genome editing trials are deployed in patients.

# Methods

## Ethics
This work was conducted in compliance with all relevant ethical regulations with full approval from the Stanford University Institutional Review Board (IRB) committee. Umbilical cord blood HSPC donors provided informed consent according to Stanford University's IRB committee (protocol # 33813) and patient information was de-identified prior to laboratory experiments—we therefore are unable to make a statement speaking to sex or ethnicity of participants. Donors were not aware of the research purpose or compensated for their participation. Consent forms permitted publication of de-identified genetic information.

## Culturing of HSPCs
Primary human HSPCs were sourced from fresh umbilical cord blood (generously provided by Binns Family program for Cord Blood Research) under protocol 33818, which was approved and renewed annually by the NHLBI IRB. All patients provided informed consent for the study. CD34[+] HSPCs were bead-enriched using Human CD34 Microbead Kits (Mitenyi Biotec, Inc., Bergisch Gladbach, Germany) according to manufacturer's protocol and cultured at $1 \times 10^5$ cells/mL in CellGenix GMP SCGM serum-free base media (Sartorius CellGenix GmbH, Freiburg, Germany) supplemented with stem cell factor (SCF) (100 ng/mL), thrombopoietin (TPO)(100 ng/mL), FLT3–ligand (100 ng/mL), IL-6 (100 ng/mL), UM171 (35 nM), 20 mg/mL streptomycin, and 20 U/mL penicillin. The cell incubator conditions were 37 °C, 5% $CO_2$, and 5% $O_2$.

## Genome editing of HSPCs
Chemically modified gRNAs used to edit HSPCs were purchased from Synthego (Menlo Park, CA, USA). The gRNA modifications added were the 2′-O-methyl-3′-phosphorothioate at the three terminal nucleotides of the 5′ and 3′ ends[37]. All Cas9 protein (SpyFi S.p. Cas9 nuclease) was purchased from Aldevron, LLC (Fargo, North Dakota, USA). The RNPs were complexed at a Cas9:sgRNA molar ratio of 1:2.5 at 25 °C for 10 min prior to electroporation. HSPCs were resuspended in P3 buffer (Lonza, Basel, Switzerland) with complexed RNPs and electroporated using the Lonza 4D Nucleofector (program DZ-100). Cells were plated at $1 \times 10^5$ cells/mL following electroporation in the cytokine-supplemented media described above.

## TSO500 library preparation
Input DNA concentration was determined by Qubit dsDNA HS assay kit on the Qubit Fluorometer according to the manufacturing protocol (Qubit, London, UK). DNA was then fragmented to 90 to 250 bp by sonication using a Covaris E220 Evolution Sonicator (Covaris, Woburn, MA, USA), with a target peak of around 130 bp as determined by Agilent Technologies 2100 Bioanalyzer using a High Sensitivity DNA chip. Samples then underwent end repair and A-tailing. Adapters containing UMIs were ligated to the ends of the DNA fragments. After a purification step, the DNA fragments were amplified using primers to add index sequences for sample multiplexing (required for cluster generation). Two hybridization/capture steps were performed. First, a pool of oligos specific to the 523 genes targeted by TSO500 with supplementary probes from Integrated DNA Technologies, Inc. (Coralville, IA, USA) (Supplementary Table 3) were hybridized to the prepared DNA libraries overnight. Next, streptavidin magnetic beads were used to capture probes hybridized to targeted regions. The hybridization and capture steps were repeated using enriched DNA libraries to ensure high specificity for the captured regions. Primers were used to amplify enriched libraries using sample purification beads. Enriched libraries were quantified and each library was normalized to ensure a uniform representation in the pooled libraries. Finally, libraries were pooled, denatured, and diluted to the appropriate loading concentration and sequenced on an Illumina NovaSeq with a read length of 2 × 151 base pairs. Up to 8 TSO500 libraries were sequenced per run.

## Indel frequency analysis by TIDE
2-4d post-targeting, HSPCs were harvested and a Qiagen DNeasy Blood & Tissue Kit (Redwood City, CA, USA) was used to collect gDNA. The following primers were then used to amplify respective cut sites with Phusion Green Hot Start II High-Fidelity PCR Master Mix (Thermo Fisher Scientific, Waltham, MA, USA) according to manufacturer's instructions: AAVS1, forward: 5′-AGGATCCTCTCTGGCTCCAT-3′, reverse: 5′-CCCCTGTCATGGCATCTTC-3′; *HBB*, forward: 5′-AGGGTT GGCCAATCTACTCC-3′, reverse: 5′-AGTCAGTGCCTATCAGAAACCCAA GAG-3′; *ZFPM2*, forward: 5′-GCAAATGCAGCAGTAGACCA-3′, reverse: 5′-CCTTCGCTCTCAATTTTGCT-3′; and *EZH2* (*ZFPM2* OT1), forward: 5′-AAAAGAGAAAGAAGAAACTAAGCCCTA-3′, reverse: 5′-TTTTCCTCCCC TCATTTCAA-3′. PCR reactions were then run on a 1% agarose gel and appropriate bands were cut and gel-extracted using a GeneJET Gel Extraction Kit (Thermo Fisher Scientific, Waltham, MA, USA) according to manufacturer's instructions. Gel-extracted amplicons were then Sanger sequenced with the forward and reverse amplicon primers shown above. Resulting Sanger chromatograms were then used as input for indel frequency analysis by TIDE (version 3.3.0)[25].

## Library preparation and sequencing
The Illumina DNA Prep (Cat. No. 20025519) and Illumina DNA Prep with Enrichment (Cat. No. 20025523) kits were used to prepare WGS and WES libraries, respectively, from 100 ng of gDNA input each, according to the manufacturer's instructions. For the DNA Prep, library quality was confirmed using the Agilent 2100 Bioanalyzer. For the DNA Prep

with Enrichment, libraries were quantified individually and quality was confirmed using the Agilent 2100 Bioanalyzer. The Illumina CEX panel (Cat. No. 20020183) was used for exome enrichment and the final hybridization time was extended to 16hrs. Libraries were denatured and diluted for sequencing on the NovaSeq 6000 according to the manufacturer's instructions. Each WGS library was diluted to 250pM and run on a single S4 flowcell, and each WES library was diluted to 100-200pM and run on a single S1 flowcell according to the manufacturer's instructions.

### Data analysis
WGS data was processed using the Illumina DRAGEN Germline pipeline v3.8.4 with default settings. Due to the volume of data, each lane was processed individually and the resulting variant call format (vcf) files were merged for variant analysis. WES data was processed using the DRAGEN Enrichment pipeline v3.8.4 with default settings. Additionally, each pipeline was used to process the Mock and Edited samples as "tumor/normal" and "normal/tumor" pairs. Known systematic noise filters were applied to all called variants. Data visualization was done in Prism (version 9) and RStudio (Version 1.2.5033).

### Reporting summary
Further information on research design is available in the Nature Research Reporting Summary linked to this article.

## Data availability
High-throughput sequencing data generated for TSO500 panel, WES, and WGS Cas9 has been uploaded to the NCBI Sequence Read Archive (SRA) submission: SRP387090 and NCBI BioProject PRJNA860159. The filtered data for all figures in this study are provided in the Supplementary Information. Source data are provided with this paper.

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

## Acknowledgements

The authors would like to thank funding from the Chan Zuckerberg Biohub that made this work possible.

## Author contributions

M.H.P. supervised the project. M.K.C., V.V.B., and M.H.P. designed experiments. M.K.C., V.V.B., E.J., M.W., J.P.H., F.C., D.K., I.K., A.G., and T.T. carried out experiments. M.K.C., V.V.B., and M.H.P. wrote the manuscript.

## Competing interests

The authors of this study declare the following competing interests: M.H.P. is on the Board of Directors of Graphite Bio. M.H.P. serves on the SAB of Allogene Tx and is an advisor to Versant Ventures. M.H.P., M.K.C., and V.V.B. hold equity in Graphite Bio. V.V.B. serves on the Board of Directors and SAB of Umoja Biopharma, the Board of Directors of ArsenalBio, and is a board observer at Synthego. M.H.P. holds equity in CRISPR Tx. E.J., M.W., F.C., I.K., A. G., and T.T. are employees of Illumina, Inc. The remaining authors declare no competing interests.
