## [Peer Review File · Nature Communications]

Reviewers' Comments:

Reviewer #1:

Remarks to the Author:

Cromer and Barsan et al., present in this manuscript a novel application of a whole-exome next generation sequencing (NGS) – based approach already used for oncotyping purposes to discover off-target editing in known coding areas of genes associated with tumorigenesis, following high fidelity Cas9 and sgRNA ribonucleoprotein (RNP) delivery in hematopoietic stem and progenitor cells (HSPCs). The innovation is the application of this workflow to detect low frequency off-target editing that could potentially trigger an oncogenic process in the engineered cells. This is a very important topic of high general interest and has particular importance in therapeutic approaches and clinical trials using CRISPR technology. The paper is well organized and well written, and could have high impact in the field. This impact will be further improved if the authors could extend their work according to the following suggestions:

1. In this project, the authors used only one cell-line: HSPCs. These are cells of mesodermal origin that have been used already to a great extent in CRISPR-based cell therapies in oncology and beyond (Finck A., et al., *Nature Communications*, 2020; Wu Y., et al., *Nature Medicine*, 2019). Since cell and gene therapies are expanding in the treatment of additional diseases, different cell types have been targeted including cells of mesenchymal and mesodermal origin (e.g. adipocytes, Tsagkaraki et al *Nat Comm* 2021) but also neuronal cells, and hepatocytes deriving from ectodermal and endodermal origin, respectively. It would be therefore interesting to enrich this study with another cell type with distinct origin and potentially chromatin behavior to HSPCs.
2. This work was performed on cells that derived from healthy donors. As the ex vivo CRISPR editing could be potentially used in patients with genetic syndromes that impair genome stability and DNA repair mechanisms, it would be interesting to see how the potential oncogenic off-target editing would be in cells bearing -to begin with- such mutations (e.g. Lynch syndrome).
3. In this study, the exons of known genes associated with tumorigenesis are screened for off-target editing. However, there have been described also intronic mutations that could lead into the development and progression of disease and cancer (Vaz-Drago et al., *Hum Genet*, 2017). How could this approach be expanded to include screening for intronic yet potentially harmful off-target editing?
4. The authors used high fidelity SpyCas9 for the gene editing. It would be interesting- as proof of principle- to also include samples transfected with a less specific Cas9 and report how the potentially tumorigenic off-target editing differs for a given target locus (eg. ZFPM2).
5. Off-target editing at EZH2 was found to be on average 19.3%. However, as shown in Fig. 1C the on-target editing of ZFPM2 is very low (ranging 13-40%). It would be very important for this study to further optimize the on-target efficiency and then reassess the level of off-target editing with this specific sgRNA. This would seem to be a necessary addition to the paper to reinforce the point.

Minor:

Main Figures 3D, 4D are not referred in the manuscript.

Reviewer #2:

Remarks to the Author:

In the manuscript entitled "Ultra-deep sequencing reveals no evidence of oncogenic mutations or enrichment by ex vivo CRISPR/Cas9 genome editing in human hematopoietic stem and progenitor cells" Cromer et al. developed a novel tumor suppressor/oncogene ultra-deep sequencing pipeline to determine whether hematopoietic stem progenitor cell (HSPC) editing and short-term ex vivo expansion leads to disruption and/or enrichment of cancer-associated variants when delivered in a clinically relevant context.

Whereas whole genome sequencing is the only way to capture variants/abnormalities across the entire genome, read depth per base pair to achieve high sensitivity is not technically feasible' Therefore, the authors limit sequencing to exons of the most common tumor suppressors and oncogenes(523 genes) to detect extremely low frequency events and they claim that no variants was observed using 3 different gRNA.

The novelty of the paper is quite limited, since they use an already established list of genes (TSO500) and they already reported in human primary cells that Cas9 RNP delivery did not introduce mutations in p53 or 129 other cancer related genes (using the Stanford Solid Tumor Actionable Mutation Panel) (Vaidyanathan, 2021; Gomez-Ospina, 2019). Interestingly, this observation is in contrast with previous publications, reporting the enrichment of p53-inactivating mutations following CRISPR-based editing in immortalized human cell lines. Unfortunately, the authors limit to the discussion the potential explanations for this difference and do not provide any experimental data to shed light on this incongruence.

Finally, the authors limit their analysis and discussion of CRISPR on-target induced genomic variation, one of the most important genotoxicity issue associated with CRISPR editing, to one sentence:

“In addition, while chromothripsis was recently reported as a rare consequence of on-target Cas9 cleavage, in our bulk population of HSPCs we found no apparent drop in read depth in variants proximal to the intended cut site for any Cas9 treatment.”

No reference to chromosome truncation is done (Cullot 2019; Boutin 2021).

Dear Nature Communications Editorial Staff,

We appreciate the time and effort that you and the reviewers have dedicated to providing your valuable feedback on our manuscript. We have incorporated changes to reflect the suggestions provided by both reviewers and have itemized a point-by-point response to the comments and suggestions. We again thank the reviewers for their insightful comments that help further improve our manuscript.

Reviewer #1:

Cromer and Barsan et al., present in this manuscript a novel application of a whole-exome next generation sequencing (NGS) – based approach already used for oncotyping purposes to discover off-target editing in known coding areas of genes associated with tumorigenesis, following high fidelity Cas9 and sgRNA ribonucleoprotein (RNP) delivery in hematopoietic stem and progenitor cells (HSPCs). The innovation is the application of this workflow to detect low frequency off-target editing that could potentially trigger an oncogenic process in the engineered cells. This is a very important topic of high general interest and has particular importance in therapeutic approaches and clinical trials using CRISPR technology. The paper is well organized and well written and could have high impact in the field. This impact will be further improved if the authors could extend their work according to the following suggestions:

Comment 1: In this project, the authors used only one cell-line: HSPCs. These are cells of mesodermal origin that have been used already to a great extent in CRISPR-based cell therapies in oncology and beyond (Finck A., et al., Nature Communications, 2020; Wu Y., et al., Nature Medicine, 2019). Since cell and gene therapies are expanding in the treatment of additional diseases, different cell types have been targeted including cells of mesenchymal and mesodermal origin (e.g. adipocytes, Tsagkaraki et al Nat Comm 2021) but also neuronal cells, and hepatocytes deriving from ectodermal and endodermal origin, respectively. It would be therefore interesting to enrich this study with another cell type with distinct origin and potentially chromatin behavior to HSPCs.

We support that future work in other types should be performed to test the reviewer’s hypothesis that there might be variability between cell types (whether because of differences in chromatin, DNA repair variability or other reasons). We believe that work goes beyond the proof-of-concept (POC) study shown here but note that the platform can be easily used in other cell types by others if they desire. In contrast to other POC studies listed below, however, we specifically chose a *clinically* relevant cell type (hematopoietic stem and progenitor cells (HSPCs), from healthy donors to avoid the limitations associated with cell lines. In this way we avoid the challenges of using cell lines with known defects in DNA damage repair and also utilize the most common stem cell type currently being used in clinical genome editing applications. Similar POC studies in high-impact journals included data primarily in cell lines, and only recently primary human cells:

Method	Journal	Year	Cas9	Delivery method	Cell types
CHANGE-Seq	Nature Biotechnology	2020	WT	Cell-free RNP	human primary T cells, HEK293, & U2OS cell lines
CIRCLE-Seq	Nature Methods	2017	WT	Cell-free RNP	HEK293, U2OS, & PGP1 cell lines
DISCOVER-Seq	Science	2019	WT	RNP electroporated into cells	K562, human iPSC, & murine B16-F10 cell lines
GUIDE-Seq	Nature Biotechnology	2015	WT	RNP electroporated into cells	HEK293 cell line
SITE-Seq	Nature Methods	2017	WT	Cell-free RNP	HEK293, K562, & HeLa cell lines

Considering the above studies published in high impact journals, we believe our POC study in primary human HSPCs is substantively more *clinically* relevant than studies performed solely in immortalized and cancer cell lines, which are known to possess numerous genomic aberrations with dysregulated DNA damage repair processes. In addition, most of these studies were conducted by delivering Cas9 RNP to cell-free genomic DNA, which has little clinical relevance to the dynamics of Cas9 activity in living cells with an intact nucleus and functional DNA damage repair mechanisms. Moreover, our selection of 3 separate human donors (deeply sequenced through a clinical oncology assay as technical triplicates) surreptitiously discovered that even one homozygous SNP in an otherwise healthy donor abrogated intended genome editing—underscoring the need to consider the genomic diversity of intent-to-treat patient populations whose haplotypes can confer additional variability beyond the cell type targeted for editing. In this example, the off-target site dropped out because of the SNP, in contrast to the creation of a new off-target site.

We have added to the discussion the other cell types and citations considered by the reviewer that we anticipate will also become more prominent in clinical trials in coming years. The specific statement added to the Discussion is as follows: “As cell and gene therapies expand in the treatment of additional diseases, an important extension of this work will be to validate our findings in additional cell types and chromatin states beyond HSPCs⁴⁴⁻⁴⁶.”

Comment 2. This work was performed on cells that derived from healthy donors. As the ex vivo CRISPR editing could be potentially used in patients with genetic syndromes that impair genome stability and DNA repair mechanisms, it would be interesting to see how the potential oncogenic off-target editing would be in cells bearing -to begin with- such mutations (e.g. Lynch syndrome)

We agree that patients with genetic syndromes that cause genomic instability (of which there are numerous) and even the effects of older age on genome integrity (e.g., clonal hematopoiesis of indeterminate potential), should be considered in broadening our understanding of where and when the CRISPR/Cas system may introduce unintended off-target edits. Our POC approach, in 3 separate donors and technical triplicates, demonstrates reproducibility and offers a workflow for application on myriad additional cell types as well as disease states. We have added to the discussion additional considerations that are highly relevant to these patient populations with cancer predisposition syndromes, oncogenic environmental exposures such as smoking/radiation, older age, immunocompromised status, etc. The specific statement added to the Discussion is as follows: “We anticipate that our approach can be expanded to additional patient populations with cancer predisposition syndromes, oncogenic environmental exposures such as smoking/radiation/bone marrow stress, older age, and immunocompromised status to validate the intrinsic versus genotoxic effects of CRISPR/Cas editing.”

Comment 3: In this study, the exons of known genes associated with tumorigenesis are screened for off-target editing. However, there have been described also intronic mutations that could lead into the development and progression of disease and cancer (Vaz-Drago et al., Hum Genet, 2017). How could this approach be expanded to include screening for intronic yet potentially harmful off-target editing?

Whereas introns are not routinely surveyed in the pathologist’s evaluation of tumors undergoing high sensitivity hybrid-capture NGS to report the variant allele frequencies of known cancer mutations, we agree introns may pose relevance to the development and progression of disease. As we discuss, however, is that current sequencing methods generally ask for their to be a compromise between breadth vs depth. The TSO500 panel has been identified by the field as representing the areas of the genome of

the greatest importance in assessing mutation spectrum for cancer. As sequencing technologies develop further, greater breadth while maintaining the same depth can be achieved. As an example of pushing the limits of this concept, we describe the following addition to the manuscript:

Our approach could be expanded either by tiling introns with additional probes (to maintain equivalent high sensitivity; particularly across introns harboring known disease-associated variants) or through whole genome sequencing. Both approaches entail significant additional sequencing cost to support high sensitivity for detecting ultra-low frequency variants. To survey introns in this study, however, we have further included deep whole genome sequencing as an additional experiment, and the data is summarized in the following main and supplementary figures:

Figure 5: Variants identified by whole-genome sequencing.

- VAF x Coverage for all variants called by WGS pipeline in AAVS1 d3 treatment with Mock d3 as background input. Large white points depict on-target AAVS1 variants.
- AAVS1 read depth x Mock read depth for all called variants in AAVS1 treatment.
- VAF x Coverage for all variants depicted in panel A after filtering. Large white points depict on-target variants.
- AAVS1 VAF x Mock VAF for all called variants in AAVS1 treatment. Large white points depict on-target variants.
- VAF x Coverage for all variants depicted in panel A after subtraction of Mock VAF from AAVS1 VAF. Large white points depict on-target variants.
- Mock VAF x AAVS1 VAF for all called variants in Mock treatment (i.e., in Mock treatment with AAVS1 as background input).

Supplemental Figure 7: Variants identified in Mock treatment by whole genome sequencing.

- a) VAF x Coverage for all variants called by whole genome sequencing pipeline in Mock d3 treatment with AAVS1 d3 as background input.
- b) Mock read depth x AAVS1 read depth for all called variants in Mock treatment.
- c) VAF x Coverage for all variants depicted in panel A after subtraction of AAVS1 VAF from Mock VAF.
- d) VAF x Coverage for all variants depicted in panel A after filtering.

Furthermore, variants remaining after filtering in AAVS1 treatment (specific methodology described in following text) will be included as an Extended Data supplement. We have also added details for this experiment to our Methods section as well as the following text to the Results:

“While intronic and intergenic variants are associated with disease progression, these are often excluded from clinical genetic workflows due to (1) difficulty in interpretation and limited clinical actionability of non-coding variants and (2) inability to achieve equivalent high coverage across the majority of the non-coding genome compared to that possible with exon capture or other targeted arrays. However, in order to determine whether *ex vivo* CRISPR-mediated genome editing introduced variants that may have been missed by both the TSO500 panel as well as whole exome sequencing, we performed whole genome sequencing on gDNA harvested from an HSPC donor at d3 post-editing with our AAVS1

gRNA (as well as Mock sample for input control). Using an established bioinformatic pipeline, we then called somatic variants present in the AAVS1 sample that were not found in the Mock control, identifying a total of 26,673 variants (mean 154 reads across called variants)(Fig. 5A & B). When plotting VAF vs. coverage depth, we observe that virtually all high-VAF variants were found at low read depth, indicating that these are likely not real variants unique to AAVS1 treatment but rather germline variants that were not sequenced at sufficient depth to be detected in the Mock sample as well. This hypothesis is supported by plotting coverage in AAVS1 vs. coverage in Mock treatment for all called variants, which reveals a linear correlation between coverage across both Mock and AAVS1 samples (Fig. 5D). We observe a clustering of low-coverage variants at VAFs of 1 and 0.5 (Fig. 5A & B), reinforcing the notion that these are germline homozygous or heterozygous variants that were also present in the Mock treatment but not found due to low coverage. Next, to remove genotype background variants, we subtracted variants detected in the Mock condition from AAVS1 variants (Fig. 5C) and again found that virtually all high VAF variants were sequenced at low coverage, likely indicating that these variants represent sequencing noise below the limit of detection. We further filtered these variants to remove calls below 0.1% VAF in the AAVS1 treatment and variants >1% VAF in the Mock sample—leaving a total 173 variants (Fig. 5E). Of these, 19 variants resided at the predicted on-target site for the AAVS1 gRNA, comprising all the highest VAF calls at coverage >100. Of these remaining candidate off-target variants (Extended Data), we found no sites with >50% homology to the AAVS1 gRNA, likely indicating false positives due to low coverage in the AAVS1 and/or Mock treatments.

As further evidence that the mutations found in the AAVS1 treatment represent sequencing artefacts, we performed additional analysis where the Mock sample was treated as the “edited” condition and the AAVS1 sample was treated as the background control. While we expect few if any true variants to be present in the Mock that were not also present in the AAVS1 treatment, we found a total of 41,200 “somatic” variants in the Mock (mean 174 reads across called variants) (Fig. 5F & Supplemental Fig. 7A-C). This list was further reduced to 171 variants after performing analogous filtering as before (Supplemental Fig. 7D). Because a nearly identical number of mutations were found in the Mock as in the AAVS1 treatment, as well as the lack of gRNA homology to any of off-target sites in the AAVS1 treatment, we believe that these data indicate no evidence of additional off-target variants introduced by CRISPR-mediated targeting at AAVS1 at the given coverage depth.”

Comment 4: The authors used high fidelity SpyCas9 for the gene editing. It would be interesting- as proof of principle- to also include samples transfected with a less specific Cas9 and report how the potentially tumorigenic off-target editing differs for a given target locus (eg. ZFPM2).

As part of another project, our group recently compared WT Cas9 with the high-fidelity variant reported in Vakulskas et al, Nature Medicine, 2018. Whereas that study originally reported a 20x reduction in off-target activity, we have more recently observed >35x reduction in activity at the 2 most prominent off-target sites for AAVS1 and *HBB* gRNAs when editing with HiFi-Cas9 compared to WT (unpublished data; shown below). We expect the paper with this comparison to be out soon and can reference it accordingly in the final manuscript.

Our group has also taken the approach to utilize the best current reagents as we move forward rather than looking backward using reagents that are less specific. It is for this reason, that we focused on this version of Cas9 (a version which is being used in the only two clinical trials that are directly correcting the disease causing mutation for sickle cell disease).

For this project we have chosen a *clinically* relevant high-fidelity SpyCas9 to elucidate performance in human cells. While we appreciate that numerous other Cas enzymes are being considered in genome editing methods (in fact, an additional high-fidelity Cas9 was reported in the past few weeks, and many more will likely be reported in the coming years), we intended this work to define a proof-of-concept approach that can be extended in the future to additional cell types, guides, Cas enzymes, and genomic loci. We have further recapped these considerations in the Discussion.

Comment 5: Off-target editing at EZH2 was found to be on average 19.3%. However, as shown in Fig. 1C the on-target editing of ZFP2 is very low (ranging 13-40%). It would be very important for this study to further optimize the on-target efficiency and then reassess the level of off-target editing with this specific sgRNA. This would seem to be a necessary addition to the paper to reinforce the point.

As we have seen with a concurrent study, gRNAs with a large number of predicted off-target sites typically display low on-target activity attributable to saturation kinetics (as is the case with VEGFA-site 1 gRNA in the forthcoming study referenced above). We believe the *ZFP2* guide elicits a similar phenomenon—i.e., with so many predicted off-target sites, our ability to achieve high frequency on-target activity is consequently limited. For relative comparison toward this point, we ran our standard *in silico* off-target prediction workflow (COSMID with standard settings) and found the following:

- AAVS1 has 5 predicted off-target sites (3 of these have a score <5, indicating high likelihood)
- HBB has 2 predicted off-target sites (both are high likelihood)
- ZFPM2 has 1,1810 sites (770 of these are high likelihood)

Optimizing the ZFPM2 gRNA is unnecessary since, unlike AAVS1 and HBB, it has little clinical relevance. Instead, this gRNA was chosen because of its extreme off-target profile to serve as a positive control to ensure our panel could pick up bona fide off-target activity should it occur. Our proposed safety workflow thus sensitively detects instances of genotoxicity in clinically relevant contexts.

Minor Comment: Main Figures 3D, 4D are not referred in the manuscript.

Thank you for catching this omission—we have fixed this.

Reviewer #2:

Comment 1: In the manuscript entitled “Ultra-deep sequencing reveals no evidence of oncogenic mutations or enrichment by ex vivo CRISPR/Cas9 genome editing in human hematopoietic stem and progenitor cells” Cromer et al. developed a novel tumor suppressor/oncogene ultra-deep sequencing pipeline to determine whether hematopoietic stem progenitor cell (HSPC) editing and short-term ex vivo expansion leads to disruption and/or enrichment of cancer-associated variants when delivered in a clinically relevant context. Whereas whole genome sequencing is the only way to capture variants/abnormalities across the entire genome, read depth per base pair to achieve high sensitivity is not technically feasible’ Therefore, the authors limit sequencing to exons of the most common tumor suppressors and oncogenes (523 genes) to detect extremely low frequency events and they claim that no variants was observed using 3 different gRNA. The novelty of the paper is quite limited, since they use an already established list of genes (TSO500) and they already reported in human primary cells that Cas9 RNP delivery did not introduce mutations in p53 or 129 other cancer related genes (using the Stanford Solid Tumor Actionable Mutation Panel) (Vaidyanathan, 2021; Gomez-Ospina, 2019). Interestingly, this observation is in contrast with previous publications, reporting the enrichment of p53-inactivating mutations following CRISPR-based editing in immortalized human cell lines. Unfortunately, the authors limit to the discussion the potential explanations for this difference and do not provide any experimental data to shed light on this incongruence.

We agree with the reviewer’s point and, indeed, a major goal of our study was to determine the likelihood of introduction of p53 or other tumor-related variants because of CRISPR-mediated editing. We believe that the incongruity with these previous publications (i.e., that we do not find any evidence of tumorigenic variants introduced or enriched following genome editing) is a function of our editing that was conducted in as clinical a context as possible (transient delivery of Cas9 RNP by electroporation to primary human HSPCs) compared to the experimental design of the prior studies which report enrichment of p53-inactivating mutations based upon: 1) the presence of p53 mutations in the initial pool of cells prior to (not as a consequence of) Cas9 delivery, which would *not* be expected to occur in primary cells derived from healthy donors; 2) stably integrated Cas9 expressed by a strong, constitutive promoter, which reliably invoke a dramatic DNA damage response; and 3) immortalized cell lines that typically have gross chromosomal abnormalities (polyploidy, aneuploidy, translocations, etc.) with dysfunctional DNA damage and nucleic acid delivery-sensing responses. We have referenced these studies in our manuscript and underscore the fundamental difference between associating *enrichment* of p53-inactivating

mutations in non-clinical experimental designs versus iatrogenic introduction of p53 mutations in a clinical context which was not observed in our ultra-deep sequencing approach. We believe these experiments will help re-orient the field to the precise results of the prior p53 experiments rather than the broad generalization that has mis-interpreted the results. We note that enrichment of cells with p53 mutations in a population of cells would occur regardless of whether CRISPR was used or not. There are multiple publications describing this effect. We believe that not seeing a detectable generation of mutations in p53 or any of the other genes on this panel by the entire manufacturing process will be of great interest to researchers, clinicians, and regulators.

Regarding the Stanford Actionable Mutation Panel (STAMP) run at our university and previously published by our colleagues, we note this hybrid capture panel only goes down to a limit of detection of 5% VAF which is precisely why we placed much greater importance to sequence more broadly and deeply to determine whether Cas9 introduced oncogenic mutations—especially since Cas9 HiFi typically elicits off-target activity well below 5% at even top off-target loci when paired with clinically relevant guides. In this work, we have achieved a 50-fold greater depth across a ~4-fold greater number of genes (so meaningful and important increases in both depth and breadth). This detail has been added to the discussion.

Comment 2: Finally, the authors limit their analysis and discussion of CRISPR on-target induced genomic variation, one of the most important genotoxicity issue associated with CRISPR editing, to one sentence: "In addition, while chromothripsis was recently reported as a rare consequence of on-target Cas9 cleavage, in our bulk population of HSPCs we found no apparent drop in read depth in variants proximal to the intended cut site for any Cas9 treatment."

We have included WES and WGS data to assess for evidence of rearrangements (particularly on the chromosome with the on-target edits (and *EZH2* off-target edit too, since it had such a high frequency)). Note, WGS data has been added to the revised manuscript as full main and supplemental figures as well as accompanying text in the Results section of the manuscript (detailed in full in above response to Reviewer 1). To check for rearrangements/chromothripsis in these datasets, we investigated discarded (unmapped) reads on chromosome of on-targets as well as *EZH2* off-target in both WES and WGS results to see if we can detect any evidence of partially mismatched reads. Toward this end, we found no detectable evidence of translocations, inversions, or chromothripsis in our bulk samples, despite sequencing at extremely high coverage depth with the TSO500 and WES workflows. While concerning, chromothripsis is expected to happen very infrequently and was only observed when via high throughput single-cell sequencing data, which is likely why we see no evidence of this in any of our bulk NGS data.

Comment 3: No reference to chromosome truncation is done (Cullot 2019; Boutin 2021).

We have added these helpful references, thank you.

We value the vote of confidence and the reviewers' time for providing thoughtful feedback on our work. We believe the manuscript is much improved with the incorporation of these changes as well as the comprehensive WGS analysis that we have added to this revised version. We hope these additions have addressed the reviewers' suggestions and look forward to your response.

Reviewers' Comments:

Reviewer #1:

Remarks to the Author:

The manuscript is suitably revised...

Reviewer #2:

Remarks to the Author:

In the revised version of the manuscript the authors:

- performed WGS of samples treated with one gRNA in order to evaluate if genomic alteration would be visible outside the coding sequences.
- ameliorated the discussion by adding references and discussing additional crucial points

However, test on additional primary cells to evaluate CRISPR safety and test on cell line to better assess the different p53 results would have been a clear and interesting addition.

The difference in the number of analysed genes and in the read depth compared to previous report is relevant, but not novel or unexpected.

Please, consider that highlighting the changes added to the novel version of the manuscript would help the reviewers in their evaluation.

Dear Nature Communications Editorial Staff,

We appreciate the time and effort that you and the reviewers have dedicated to providing valuable feedback on our manuscript. We have incorporated changes to reflect the suggestions provided by both reviewers and have itemized a point-by-point response to the comments and suggestions. We again thank the reviewers for their insightful comments that helped further improve our manuscript for publication in Nature Communications.

REVIEWERS' COMMENTS

Reviewer #1 (Remarks to the Author):

The manuscript is suitably revised...

Reviewer #2 (Remarks to the Author):

In the revised version of the manuscript the authors:

- performed WGS of samples treated with one gRNA in order to evaluate if genomic alteration would be visible outside the coding sequences.*
- ameliorated the discussion by adding references and discussing additional crucial points*

However, test on additional primary cells to evaluate CRISPR safety and test on cell line to better assess the different p53 results would have been a clear and interesting addition.

We absolutely agree that future work should be performed to determine the impact of germline vs somatic variability on CRISPR safety across cell types (whether because of differences in chromatin, DNA repair variability, mutated p53, or other reasons). We have added such considerations to the discussion and underscore that our proof-of-concept clinical study was intended to specifically evaluate a *clinically* relevant cell type (hematopoietic stem and progenitor cell) across healthy donors to avoid the limitations associated with cell lines. Specifically, we avoided the challenges of using cell lines with known defects in DNA damage repair and instead focused our efforts on the most common stem cell type currently being used in clinical genome editing applications. Whereas additional cell types extend beyond the clinical POC study shown here, we acknowledge the platform may be readily adapted to determine genotoxicity / mutation burdens in a wide variety of contexts through follow up studies. We have included in the discussion accordingly, "As cell and gene therapies expand in the treatment of additional diseases, an important extension of this work will be to validate our findings in additional cell types and chromatin states beyond HSPCs⁴⁴⁻⁴⁶."

The difference in the number of analyzed genes and in the read depth compared to previous report is relevant, but not novel or unexpected.

While we agree that other studies have assayed either a greater number of genes or achieved high sequencing depth, our novelty lies in applying both high-depth *and* comprehensive coverage of the most common variants across 500+ genes described in human cancers in the context of a clinically relevant ex vivo gene editing of HSPCs. Our workflow exceeds current standards of care for detecting rare variants in cancer patients, since most hybrid-capture NGS panels achieve a limit of detection of 1-5% VAF. Therefore, sequencing more broadly and deeply helped us establish whether Cas9 introduced oncogenic

mutations that may enrich from very low initial mutation frequencies—especially since Cas9 HiFi typically elicits off-target activity well below 5% at even top off-target loci when paired with clinically relevant guides. In this work, we have achieved a 50-fold greater depth across a ~4-fold greater number of genes (yielding meaningful and important increases in both depth and breadth). This detail has been added to the discussion.

We value the vote of confidence and the reviewers' time for providing thoughtful feedback on our work. We believe the manuscript is much improved with the incorporation of these changes as well as the comprehensive WGS analysis that we have added to this revised version. We hope these additions have addressed the totality of reviewers' suggestions and look forward to your response.